# Physical Fitness and Body Composition in Women with Systemic Lupus Erythematosus

**DOI:** 10.3390/medicina55020057

**Published:** 2019-02-21

**Authors:** Sergio Sola-Rodríguez, Blanca Gavilán-Carrera, José Antonio Vargas-Hitos, José Mario Sabio, Pablo Morillas-de-Laguno, Alberto Soriano-Maldonado

**Affiliations:** 1Department of Education, Faculty of Education Sciences, University of Almería, 04120 Almería, Spain; asoriano@ual.es; 2Department of Physical Education and Sport, Faculty of Sport Sciences, University of Granada, 18071 Granada, Spain; bgavilan@ugr.es (B.G.-C.); pmorillasdelaguno@hotmail.com (P.M.-d.-L.); 3Systemic Autoimmune Diseases Unit, Department of Internal Medicine, “Virgen de las Nieves” University Hospital, 18014 Granada, Spain; joseantoniovh@hotmail.com (J.A.V.-H.); jomasabio@gmail.com (J.M.S.); 4SPORT Research Group (CTS-1024), CERNEP Research Center, University of Almería, 04120 Almería, Spain

**Keywords:** physical fitness, flexibility, muscular strength, cardiorespiratory fitness, body composition, systemic lupus erythematosus, obesity

## Abstract

*Background and objectives:* Higher physical fitness is associated with a more favorable weight and body composition in the general population, although this association has not been studied in patients with systemic lupus erythematosus (SLE). The aim of the present study was to examine the association of different components of physical fitness with body composition in women with SLE with mild disease activity. *Materials and Methods:* This cross-sectional study included 77 women with SLE (43.2 ± 13.8 years old) and clinical stability during the previous 6 months. Body composition (including body mass index (BMI), fat mass index (FMI), waist circumference, waist-to-height ratio and waist-to-hip ratio) was assessed using a stadiometer, an anthropometric tape, and a bioimpedance device. Physical fitness included cardiorespiratory fitness (Siconolfi step test and 6 min walk test), muscular strength (handgrip strength test as upper body measure and 30 s chair stand as lower body measure), and flexibility (back-scratch test). Participants with a fitness level equal or above the median of the study sample were categorized as “fit” and those below the median were categorized as “unfit”. Linear regression assessed the association of physical fitness with body composition parameters. *Results:* Cardiorespiratory fitness and upper body muscular strength were negatively associated with BMI, FMI, waist circumference, and waist-to-height ratio (all, *p* < 0.05). Lower body muscular strength and flexibility were negatively related to FMI, waist circumference, waist-to-height ratio, and waist-to-hip ratio (all, *p* < 0.05). These relationships were still significant after controlling for age, disease duration, accrual damage, and SLE activity. Overall, fit patients presented significantly lower values in all body composition parameters compared to unfit patients (all, *p* < 0.05). *Conclusions:* The main findings of the present study suggest that physical fitness is inversely associated with body composition in women with SLE. Given the cross-sectional nature of this study, future clinical trials should study the causal pathways underlying these relationships.

## 1. Introduction

Systemic lupus erythematosus (SLE) is an autoimmune disease of unknown origin which affects approximately 20 of every 100,000 females [1]. It is characterized by an immune mediated damage that may affect the skin, joints, kidney, brain, and most systems and organs [2]. SLE is associated with progressive and irreversible accrual organic damage, which has been shown to be a predictor of morbidity and early mortality [3]. However, the prognosis of the disease and patients’ quality of life has improved in recent years as a result of better diagnostic methods and more effective treatments [4].

Cardiovascular diseases currently represent one of the main causes of mortality in this population [5]. Obesity, which increases the risk of cardiovascular disease and atherosclerosis, is present in nearly 50% of women with SLE [6]. In particular, body mass index (BMI; as a measure of excess of body weight), fat mass index [7,8] (FMI; as measure of excess of body fat), waist circumference, and waist-to-hip and waist-to-height ratios [9] (as measures of central fat) represent independent predictors of cardiovascular disease [10,11]. Ramírez et al. [12] observed that women with SLE present a higher BMI, waist-to-hip ratio, and FMI compared to the general population. Adiposity is intimately associated with a systemic low-grade chronic inflammatory state, thus contributing to cardiovascular disease risk in this population [13]. Therefore, identifying modifiable factors that can potentially be associated with a more favorable body weight and composition is of clinical interest.

Physical fitness is a powerful marker of present and future cardiovascular health [10] that can be modified through exercise. In the general population, higher fitness significantly attenuates the detrimental effect that obesity has on cardiovascular health and cardiovascular mortality [14]. Similarly, lower physical fitness has also been related to higher levels of adiposity and higher BMI in the general population [10]. For instance, muscular strength has been inversely related to adiposity and cardiometabolic risk [15], and cardiorespiratory fitness (CRF) has been negatively associated with BMI [16] and positively related to an increased fat-free mass in women [17].

Patients with SLE present a reduced CRF [18,19,20,21,22], functional capacity [23,24], and muscular strength [25,26], and these fitness components have been positively associated with health-related outcomes [19,20,23]. However, the association of physical fitness components with body weight and composition has not been studied in detail in this population. This information is relevant because it could lead to implementation of new studies focused at increasing fitness levels in this population with the aim of improving body composition and ultimately reducing cardiovascular risk.

The aim of the present study was to evaluate the association of different components of physical fitness (CRF, muscular strength, and flexibility) with body composition (BMI, FMI, waist circumference, waist-to-height ratio, and waist-to-hip ratio) in women with SLE.

## 2. Materials and Methods

### 2.1. Design and Participants

In this cross-sectional study, a total of 172 Caucasian patients with SLE were invited to participate in the study. Recruitment was performed through the Systemic Autoimmune Diseases Unit of the “Virgen de las Nieves” University Hospital and the “San Cecilio” University Hospital in Granada. Inclusion criteria were: Women aged between 18 and 60 years, with ≥4 SLE classification criteria provided by the American College of Rheumatology [27], a minimum follow-up of one year at our unit, and clinical stability (i.e., the absence of changes in the systematic lupus erythematosus disease activity index (SLEDAI) and/or treatment) during the previous 6 months. Exclusion criteria were: Not being able to read, understand, and/or sign the informed consent; cancer; history of clinical cardiovascular disease and/or lung disease in the last year, or receiving doses of biological treatment higher than 10 mg/d of prednisone (or equivalent) in the previous 6 months.

All participants received detailed information about the study aims and procedures and signed informed consent before being included in the study. The Research Ethics Committee of Granada reviewed and approved the study protocol on 31 October 2016 (reference number: 09/2016).

The flowchart of the participants included in this study is presented in Figure 1. From a total of 172 patients initially invited, 81 refused to participate (41 patients reported living very far from the hospital, 36 were not able to find time to perform the evaluations, and 4 were not interested), 12 patients did not present clinical stability during the 6 previous months, and 2 patients had cardiovascular disease during the previous year. A total of 77 women with SLE (mean age 43.2, SD 13.8) met the inclusion criteria, agreed to participate, and were assessed in two waves (49 women in October 2016 and 28 women in February 2017). Both evaluations were identical, with the exception that the 6 min walk test (6MWT) and Siconolfi step test were not carried out (*n* = 28) in the wave of 2017 due to timing issues. One woman did not perform the handgrip strength test and the back-scratch test due to a wrist injury.

### 2.2. Anthropometric Measures

Height (cm) was measured using a stadiometer (SECA 222, Hamburg, Germany) and weight, fat mass, and lean mass (kg) with a bioimpedance device (InBody R20, Biospace, Seoul, Korea). BMI (weight in kg/height in m^2^) and FMI (fat mass in kg/height in m^2^) were calculated. Waist perimeter and hip circumference (cm) were measured with an anthropometric tape (Harpenden, Holtain Ltd., Wales, UK). The waist-to-hip (waist circumference/hip perimeter) and waist-to-height (waist circumference/height) ratios were calculated.

### 2.3. Physical Fitness Measures

Cardiorespiratory fitness was assessed using the Siconolfi step test and the 6MWT. The Siconolfi step test [28] has been previously validated to estimate maximum oxygen consumption (VO_2max_) in patients with SLE [29]. The test was carried out as described in the original protocol [28], using a wooden box (25.4 cm high × 30.5 cm wide × 45.7 cm long). In stage 1, each patient was instructed to step up and down the box for 3 min at a rate of 17 times per min, controlled by a metronome. Heart rate (HR) was continuously monitored with an HR monitor (Polar V800, Osakeyhtiö, Kempele, Finland). The test was finalized when the average HR during the last 30 s of stage 1 reached ≥65% of the estimated maximum HR (220-age). Otherwise, the participant would perform a second stage of 3 min at a rate of 26 times per min. In case the participant did not reach ≥65% of the maximum HR during the last 30 s of stage 2, a third stage was performed at a rate of 32 times per min. The stage in which 65% of the maximum HR was reached was recorded and VO_2max_ was estimated from the following formula [28]:(1)$$\begin{array}{ll} VO2max\;= & \;0.302\;\times \;\left(stage\;multiplier\;\times \;body\;weight/1000\right)/(((0.667\\ & \;\times heart\;rate\;stage)-42)/100)-\left(0.019\;\times \;age\right)\;+\;1.593 \end{array}$$
where stage multiplier was 16.287 for stage 1, 24.910 for stage 2, and 35.533 for stage 3. Stage HR corresponds to average HR obtained during the last 30 s of the highest stage reached.

The 6MWT measures the maximum distance (in meters) that a person can walk in six min [30]. This test has been widely used in rheumatic diseases, including patients with SLE [24]. The test was performed along a 50 m circuit, broken into 5 m long sections by 10 cones. The total distance was calculated as the number of complete laps plus the number of sections covered within the last lap in case of an incomplete final lap at the expiration of the allowed time.

Muscular strength was assessed through the 30 s chair stand test (lower body) and the handgrip strength test (upper body). The 30 s chair stand test [30,31] measures the number of times a person can get up completely from a chair, starting from a sitting position, with a straight back and feet flat on the floor in 30 s. The handgrip strength test [32] was assessed using a digital dynamometer (Model T.K.K.540®; Takei Scientific Instruments Co., Ltd., Niigata, Japan). Alternating between the two hands, participants performed the test twice with a one-min break between the two attempts of each hand. The best score of each hand was used to compute an average of the two scores.

Upper body flexibility was assessed through the back-scratch test [30] that measures how close the hands can be brought together behind the back. In the standing position, the participant should place one hand (facing inwards, fingers extended) behind the head and back over the shoulder and move down the back to reach as far as possible. The other hand should be placed behind the back (palm facing outward, fingers extended) and reach up as far as possible, trying to touch or overlap the middle fingers of both hands. The distance between the tips of the middle fingers of the hands was measured. If the fingers only touch, the score would be “zero”, if they do not touch the score would be negative and if they overlap the score would be positive. The participants performed the test twice with each hand and the average of the best value from both hands was used.

### 2.4. Other Measurements

All participants filled out a sociodemographic and clinical data questionnaire to gather information, such as age, disease duration, presence of dyslipidemia, diabetes, hypertension, current medication (including antidiabetics and corticosteroids), and tobacco consumption. The systemic lupus erythematosus disease activity index (SLEDAI) was included to assess disease activity [33]. SLEDAI considers the presence or absence of several clinical and analytical manifestations. The final score goes from 0–105, where a higher score shows a higher degree of disease activity. The degree of tissue damage from the onset of the disease was evaluated by the Systemic Lupus International Collaborating Clinics/American College of Rheumatology Damage Index (SDI) [34]. The score ranges from 0–40, where a higher score means greater damage produced by SLE.

### 2.5. Statistical Analysis

The descriptive characteristics of the study sample are presented as mean and standard deviation unless otherwise indicated. The physical fitness and body composition-related variables were assessed for normality through the Kolmogorov-Smirnov Test and Q–Q charts, and all showed an approximately normal distribution. Scatter plots and Pearson’s bivariate correlations were used as preliminary analyses to understand the raw association of physical fitness (Siconolfi, 6MWT, handgrip, 30 s chair stand, and back-scratch tests) with body composition (BMI, FMI, waist circumference, waist-to-height ratio, and waist-to-hip ratio). Subsequently, linear regression models were built, including each parameter of body composition as dependent variables and each fitness test as independent variables in separate regression models along with age, SLEDAI, SDI, and disease duration as relevant factors that might confound the association of interest [35]. Additionally, participants were categorized as “fit” (i.e., equal or above the median in each fitness test) or “unfit” (below the median in each fitness test) and both groups (fit vs. unfit) were compared using analysis of covariance (ANCOVA) with the aforementioned covariables. Statistical analysis was performed with SPSS v.23 (IBM, New York, NY, USA). Statistical significance was established at *p* < 0.05 (all *p*-values were two-tailed).

## 3. Results

The descriptive characteristics of the study participants are presented in Table 1. The average BMI was 25.5 (SD 4.5) kg/m^2^, the average waist circumference was 81.7 (SD 10.7) cm, and the average hip circumference was 98.1 (SD 9.9) cm. The average VO_2max_ assessed with the Siconolfi step test was 24.4 (SD 3.2) mL/kg/min and the average distance in the 6MWT was 570.8 (SD 71.7) meters. The average handgrip strength was 23.6 (SD 5.3) kg, the average score of the 30 s chair stand test was 14.9 (SD 3.1), and the average score of the back-scratch test was −0.8 (SD 9.6) cm.

The raw association (without adjustment variables) between different components of physical fitness and BMI, FMI, waist circumference, waist-to-height ratio, and waist-to-hip ratio in women with SLE is presented in Figure 2. CRF was inversely associated with BMI, FMI, waist circumference, and waist-to-height and waist-to-hip ratios (r_range_ = from −0.74 to −0.43; all *p* < 0.05). Lower body muscular strength was inversely associated with waist circumference, and waist-to-height and waist-to-hip ratios (r_range_ = from −0.40 to −0.31; all *p* < 0.05). Flexibility was inversely associated with BMI, FMI, waist circumference, and waist-to-height and waist-to-hip ratios (r_range_ = from −0.52 to −0.38; all *p* < 0.05).

The linear regression models evaluating the association between physical fitness levels and body composition in women with SLE are presented in Table 2. Both crude and adjusted analysis yielded virtually the same results. CRF, assessed through the Siconolfi step test, was inversely associated with BMI (unstandardized coefficient (B) = −1.83; 95% confidence interval (CI) −2.36 to −1.31; *p* < 0.001), FMI (B = −1.41; 95% CI −1.91 to −0.91; *p* < 0.001), waist circumference (B = −3.73; 95% CI −4.74 to −2.71; *p* < 0.001), and waist-to-height ratio (B = −0.02; 95% CI −0.03 to −0.01; *p* < 0.001). CRF, assessed through 6MWT, was inversely associated with BMI (B = −0.04; 95% CI −0.06 to −0.02; *p* = 0.001), FMI (B = −0.04; 95% CI −0.06 to −0.02; *p* < 0.001), waist circumference (B = −0.05; 95% CI −0.10 to −0.01; *p* = 0.029), and waist-to-height ratio (B = −0.001; 95% CI −0.001 to 0.001; *p* = 0.001). The association of handgrip strength with the outcome variables was further adjusted for body weight, as some studies [36,37] have shown that obese people tend to perform best in the handgrip strength test. Muscular strength, as assessed through handgrip strength, was inversely associated with BMI (B = −0.14; 95% CI −0.24 to −0.04; *p* = 0.006), FMI (B = −0.21; 95% CI −0.31 to −0.11; *p* < 0.001), waist circumference (B = −0.40; 95% CI −0.70 to −0.11; *p* = 0.007), and waist-to-height ratio (B = −0.004; 95% CI −0.01 to −0.002; *p* = 0.001). Muscular strength, as assessed through the 30-s chair stand test, was inversely associated with waist circumference (B = −1.20; 95% CI −2.30 to −0.10; *p* = 0.033), the waist-to-height ratio (B = −0.01; 95% CI −0.01 to −0.001; *p* = 0.03), and the waist-to-hip ratio (B = −0.01; 95% CI −0.01 to −0.002; *p* = 0.011). Finally, flexibility, assessed through the back-scratch test, was inversely associated with BMI (B=-0.21; 95% CI −0.31 to −0.10; *p* < 0.001), FMI (B = −0.18; 95% CI −0.28 to −0.09; *p* < 0.001), waist circumference (B = −0.40; 95% CI −0.67 to −0.14; *p* = 0.003), waist-to-height ratio (B = −0.004; 95% CI −0.005 to −0.002; *p* < 0.001), and waist-to-hip ratio (B = −0.002; 95% CI −0.004 to −0.001; *p* < 0.001). Further adjustment for dyslipidemia, diabetes, arterial hypertension, smoking, and intake of statins and antidiabetics, or corticosteroids (and accumulated corticosteroid dose in the previous 3 years) as possible relevant confounders [34] did not change the results.

Figure 3 shows the differences in body composition between patients categorized as “fit” and those categorized as “unfit” according to the median value for each fitness test. Regarding CRF, women categorized as “unfit” (using the Siconolfi step test) had a higher BMI (mean difference 5.44 kg/m^2^; 95% CI 2.61–8.26; *p* < 0.001), FMI (mean difference 3.51 kg/m^2^; 95% CI 0.99–6.03; *p* = 0.007), waist circumference (mean difference 14.18 cm; 95% CI 8.72—19.64; *p* < 0.001), waist-to-height ratio (mean difference 0.08 units; 95% CI 0.05–0.12; *p* < 0.001), and waist-to-hip ratio (mean difference 0.05 units; 95% CI 0.02–0.09; *p* = 0.001) than those categorized as “fit”. Women categorized as “unfit” (using the 6MWT) had a higher BMI (mean difference 6.02 kg/m^2^; 95% CI 3.24–8.81; *p* < 0.001), FMI (mean difference 4.46 kg/m^2^; 95% CI 2.02–6.90; *p* = 0.001), waist circumference (mean difference 11.81 cm; 95% CI 5.70–17.96; *p* < 0.001), waist-to-height ratio (mean difference 0.09 units; 95% CI 0.05–0.12; *p* < 0.001), and waist-to-hip ratio (mean difference 0.05 units; 95% CI 0.01–0.08; *p* < 0.008) than those categorized as “fit”. Regarding strength, women categorized as “unfit” (using the handgrip strength test) had a higher BMI (mean difference 1.08 kg/m^2^; 95% CI 0.10–2.05; *p* = 0.03), FMI (mean difference 1.59 kg/m^2^; 95% CI 0.58–2.59; *p* = 0.002), waist circumference (mean difference 3.34 cm; 95% CI 0.45–6.23; *p* = 0.024), and waist-to-height ratio (mean difference 0.03 units; 95% CI 0.008–0.05; *p* = 0.008) than those categorized as “fit”. Women categorized as “unfit” (using the 30-sec chair stand test) had a higher waist-to-height ratio (mean difference 0.03 units; 95% CI 0.002–0.07; *p* = 0.041), and waist-to-hip ratio (mean difference 0.04 units; 95% CI 0.01–0.07; *p* = 0.011) than those categorized as “fit”. Regarding flexibility, women categorized as “unfit” (using the back-scratch test) had a higher BMI (mean difference 3.21 kg/m^2^; 95% CI 1.16–5.27; *p* = 0.003), FMI (mean difference 2.59 kg/m^2^; 95% CI 0.82–4.36; *p* = 0.005), waist circumference (mean difference 5.08 cm; 95% CI 0.02–10.14; *p* = 0.049), waist-to-height ratio (mean difference 0.04 units; 95% CI 0.01–0.08; *p* = 0.007), and waist-to-hip ratio (mean difference 0.03 units; 95% CI 0.005–0.06; *p* = 0.022) than those categorized as “fit”.

## 4. Discussion

The main findings of this study suggest that physical fitness is inversely associated with body weight and composition in women with SLE. These results were consistent regardless of the fitness component evaluated and despite the adjustment for multiple potential confounders.

In this study, CRF was inversely associated with BMI, FMI, waist circumference and waist-to-height ratio, although not with waist-to-hip. We observed differences in all the body composition outcomes when comparing “fit” and “unfit” patients according to the Siconolfi step and the 6MWT tests. Our results cannot be compared with other studies in SLE or similar conditions, as our objective has not been previously addressed in the literature. Our findings are, however, consistent with previous studies in the general population that showed an inverse association between CRF and body composition across all parameters. Dagan et al. [38] found negative associations of CRF with BMI and waist circumference in adult women. Other studies have also observed negative associations of CRF with total body fat [39] and central obesity (assessed by waist circumference [40], waist-to-hip ratio and waist-to-height ratio [41]). We hypothesized that an increase of CRF might be related to improvements in body composition-related parameters, since CRF substantially attenuates the obesity-related health risks through a reduction of abdominal adiposity [42] and total body fat [43]. It is worth noting that CRF is determined by genetic factors and physical activity, and it could be possible that CRF was an indirect measure of the physical activity performed, a recognized determinant of body composition in SLE [35]. On the other direction of the pathway between CRF and body composition, it is also plausible that greater fat mass leads to decreased VO_2max_ by reducing the amount of lean tissue (that extracts oxygen during the test) per kg of body weight. Importantly, CRF was the fitness component that presented the strongest inverse association with most body composition indicators assessed in the present study. The present results, along with the previous evidence relating positively CRF to different health outcomes in SLE [22,24], highlight that CRF is a relevant health marker in this particular population. Interventions aimed at increasing CRF in patients with SLE are warranted and have shown to be effective and safe for these patients [24]. For instance, Soriano-Maldonado et al. [24] recently showed that a 12-week aerobic exercise intervention combining continuous and interval sessions of progressing intensity improved the time to achieve the 85% of the maximal heart rate by an average of 2.3 min (i.e., corresponding to approximately 7.5 mL/kg/min of VO_2max_).

A negative association between handgrip muscular strength and BMI, FMI, waist circumference and waist-to-height ratio was observed, but only when adding body weight [36,37] into the models. This finding suggests that a higher isometric muscular strength of the upper limb, relative to total body weight, might be an indicator of body composition. No previous studies have focused on the association between isometric muscular strength and body composition in SLE. Hayat et al. [44] found that grip strength was positively related to BMI, but after further adjustment for weight, as we did in our study, grip strength and BMI were inversely associated. Regarding fat mass percentage (FM%), another study found that absolute grip strength was inversely associated with fat mass in elderly people [45]. In terms of central adiposity, grip strength has been negatively related to waist circumference in older adults [46]. With reference to lower-body strength, we found no association between BMI, FMI, and the 30 s chair stand test, and no differences were observed between BMI, FMI, waist circumference and 30 s chair stand test either when comparing “fit” and “unfit” individuals. The differences found between the strength tests used could be due to the different features of strength assessed. While handgrip may be representative of the maximal isometric strength [47], the functional 30 s chair stand test may be representative of the maximum strength in older adults [31] but not necessarily in adults. Indeed, previous studies in postmenopausal women did not find an association between the 30 s chair stand test and waist circumference [48], whereas there was an association between the 30 s chair stand test and BMI in older women [49]. A higher muscular strength, and its maintenance, increases resting metabolic rate, increases high density lipoprotein cholesterol (HDL-c), decreases low density lipoprotein cholesterol (LDL-c), decreases triglycerides, and increases insulin resistance, thus contributing to a better body composition [50,51]. It is also possible that central obesity could reduce muscular strength through increasing subclinical inflammation and insulin resistance. Pro-inflammatory cytokines and adipokines are produced in adipose tissue, especially in visceral areas [52], leading to catabolism, and contributing to muscle mass and strength decline [53]. Moreover, insulin resistance and a reduced anabolic action of insulin has been related to adiposity [54] and to loss of muscular strength [55]. Future prospective studies should elucidate whether muscular strength is able to counteract the effect of obesity in SLE, as it has been demonstrated in other populations [10,56].

We also found a negative relationship between flexibility and obesity in women with SLE. An inverse relationship between lower-body flexibility and BMI has been previously described in older men but not in women [57] or was not found in either gender [58]. Previous research linking fitness to cardiometabolic risk found higher values of waist circumference related to upper-body flexibility in perimenopausal women [59] and to lower-body flexibility in elderly people [60]. The low number of studies focused on assessing the benefits of flexibility in health, along with the heterogeneity of the populations and the methods used to assess it, constitute a limitation for the interpretation of the present results, which needs to be contrasted in future reports. The back-scratch test used in the present study assesses scapular mobility, and a reduced scapular motion has been previously linked to higher BMI by Gupta et al. [61]. Stretching has been shown to reduce inflammation [62] and improve vascular function [63], and it remains unknown whether these benefits could potentially be related to decreases in BMI. On the other hand, we speculate that a larger body size (especially greater arm mass) might limit the joint range of motion in this test. In line with our findings regarding FM%, the concentration of adipose tissue around the joints possibly increases the friction between the surfaces of the joints, which reduces the ability of stretching and may reduce, therefore, flexibility [64]. Due to the potential relevance of flexibility for health in SLE [65] and other populations [59,60,64], interventions and prospective studies are needed to clarify the role of this fitness component and the nature of the relationships found in this study.

This study has limitations that should be highlighted. The cross-sectional design of our study excludes establishment of causal relationships, and future prospective research should confirm or contrast these findings and attempt to evaluate the directionality of the association. The sample size was relatively small, excluding men and women with medium-to-high disease activity. Therefore, our results do not pretend to be generalizable to the whole SLE population.

## 5. Conclusions

In conclusion, the results of the present study suggest that lower physical fitness is associated with higher body weight and more unfavorable body composition (i.e., including measures of total and central adiposity) in women with SLE. More specifically, CRF, muscular strength, and flexibility components are inversely associated with BMI, FMI, waist circumference, and waist-to-height ratio. The relatively low levels of CRF and muscular strength, and the high prevalence of obesity consistently observed in this population underlie the need to take preventive actions to improve these health parameters. Due to the tight and bidirectional connection between physical fitness and body composition, further prospective and experimental research is needed to elucidate how their interaction affects the cardiovascular health of patients with SLE.

## Figures and Tables

**Figure 1 medicina-55-00057-f001:**
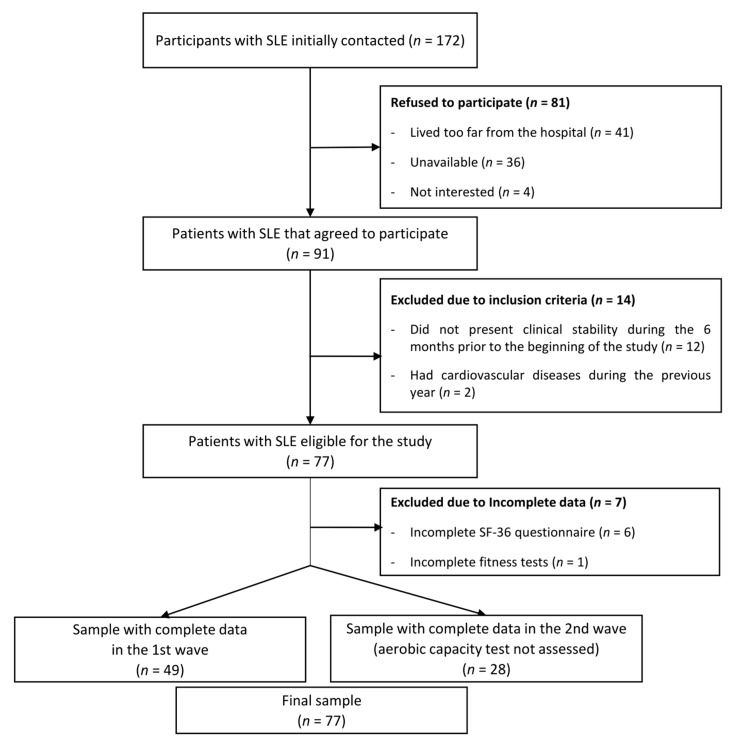
Flow diagram of the inclusion of women with systemic lupus erythematosus (SLE) for the present study.

**Figure 2 medicina-55-00057-f002:**
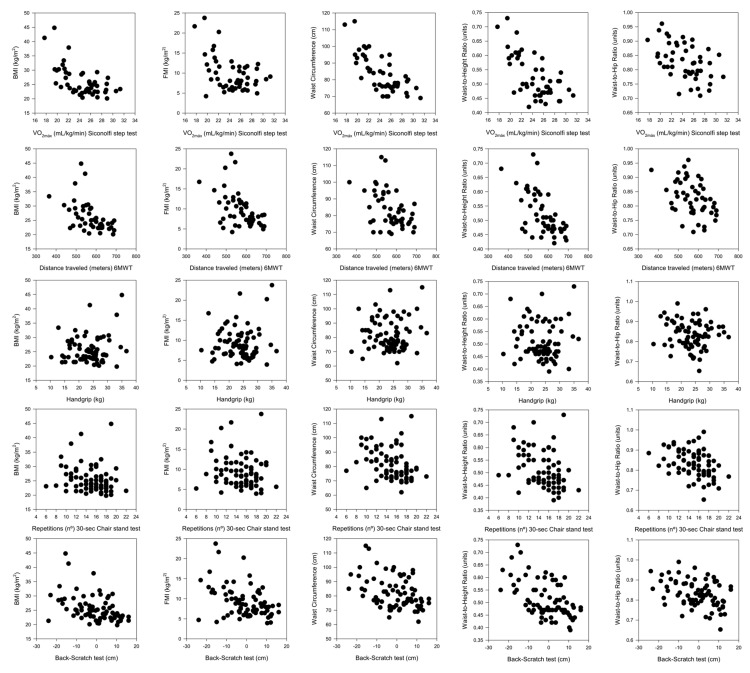
Graphic representation of the crude association between components of physical fitness and different components of body composition. 6MWT: 6-min walk test, BMI: Body mass index, FMI: Fat mass index.

**Figure 3 medicina-55-00057-f003:**
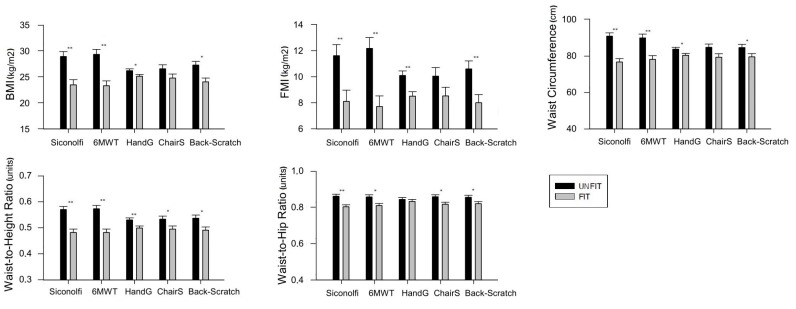
Means (95% confidence interval) of parameters of body composition in “fit” and “unfit” patients according to the median value for cardiorespiratory fitness (24.5 VO_2max_ in the Siconolfi step test; 575 m in the 6 min walk test (6MWT)), upper body strength (24.2 kg in handgrip strength test), lower body strength (15 repetitions in the 30 s chair stand test), and flexibility (1.35 cm in the back-scratch test). Differences between groups were studied using analysis of covariance (ANCOVA) with age, SLEDAI, SDI, and disease duration entered as covariates. * *p* < 0.05, ** *p* < 0.01. All analyses were adjusted for age, SLEDAI, SDI, and disease duration. Handgrip was additionally adjusted for body weight.

**Table 1 medicina-55-00057-t001:** Descriptive characteristics of the study participants.

	N	Mean	SD
Age (years)	77	43.2	13.8
Weight (kg)	77	65.1	11.1
Height (cm)	77	160.1	6.8
BMI (kg/m^2^)	77	25.5	4.5
FMI (kg/m^2^)	77	9.23	3.87
Waist circumference (cm)	77	81.7	10.7
Hip circumference (cm)	77	98.1	9.9
Waist-to-height ratio (units)	77	0.51	0.07
Waist-to-hip ratio (units)	77	0.83	0.06
Back-Scratch test (cm)	76	−0.8	9.6
Handgrip strength (kg)	76	23.6	5.3
Chair stand test (rep)	77	14.9	3.1
6MWT (m)	49	570.8	71.7
VO_2max_ estimated (mL/kg/min)	49	24.4	3.2
SLEDAI (score)	77	0.68	1.5
SDI (score)	77	0.55	1.11
Duration of SLE (years)	77	13.9	10.1
Accumulated corticosteroid dose (last 3 years; mg)	77	2875	2677
Dyslipidemia (%)	77	18	
Diabetes (%)	77	1	
Arterial hypertension (%)	77	17	
Smokers (%)	77	53	
Statins intake (%)	77	18	
Antidiabetic drugs intake (%)	77	3	
Corticosteroid dose (%; mg/d)	77	65	

FMI: Fat mass index; BMI: Body mass index; SLEDAI: Systemic lupus erythematosus disease activity index; SDI: Systemic Lupus International Collaborating Clinics/American College of Rheumatology Damage Index; 6MWT: 6 min walk test; SLE: Systemic lupus erythematosus. All variables show mean and SD values except accumulated corticosteroid dose, dyslipidemia, diabetes, arterial hypertension, smokers, statins intake, antidiabetic drugs intake, and corticosteroid dose.

**Table 2 medicina-55-00057-t002:** Linear regression analysis evaluating the association between different components of physical fitness and body composition in women with systemic lupus erythematosus *.

	β	B	SE	CI 95%		*p*
**Siconolfi VO_2max_** *						
Body Mass Index	−1.14	−1.83	0.26	−2.35	−1.30	<0.001
Fat Mass Index	−1.05	−1.41	0.24	−1.91	−0.91	<0.001
Waist Circumference	−1.09	−3.73	0.50	−4.74	−2.71	<0.001
Waist-to-Height Ratio	−0.90	−0.02	0.00	−0.02	−0.01	<0.001
Waist-to-Hip Ratio	−0.30	−0.01	0.00	−0.01	0.00	0.151
**6MWT** *						
Body Mass Index	−0.57	−0.04	0.01	−0.06	−0.01	<0.001
Fat Mass Index	−0.65	−0.03	0.00	−0.05	−0.02	<0.001
Waist Circumference	−0.35	−0.05	0.02	−0.10	−0.00	0.029
Waist-to-Height Ratio	−0.53	−0.001	0.00	−0.001	<0.001	0.001
Waist-to-Hip Ratio	−0.22	<0.00	0.00	0.00	0.00	0.162
**Handgrip**						
Body Mass Index	−0.16	−0.14	0.05	−0.24	−0.04	0.006
Fat Mass Index	−0.29	−0.21	0.05	−0.31	−0.11	<0.001
Waist Circumference	−0.19	−0.40	0.14	−0.69	−0.11	0.007
Waist-to-Height Ratio	−0.29	−0.004	0.00	−0.01	−0.00	0.001
Waist-to-Hip Ratio	−0.15	−0.002	0.00	−0.00	0.00	0.246
**Chair Stand Test**						
Body Mass Index	−0.23	−0.33	0.24	−0.81	0.14	0.168
Fat Mass Index	−0.29	−0.36	0.20	−0.77	0.04	0.083
Waist Circumference	−0.35	−1.20	0.55	−2.30	−0.09	0.033
Waist-to-Height Ratio	−0.36	−0.01	0.00	−0.01	−0.00	0.030
Waist-to-Hip Ratio	−0.41	−0.01	0.00	−0.01	−0.00	0.011
**Back-Scratch**						
Body Mass Index	−0.43	−0.20	0.05	−0.31	−0.09	<0.001
Fat Mass Index	−0.44	−0.18	0.05	−0.28	−0.09	<0.001
Waist Circumference	−0.36	−0.40	0.13	−0.66	−0.14	0.003
Waist-to-Height Ratio	−0.48	−0.004	0.00	−0.01	−0.00	<0.001
Waist-to-Hip Ratio	−0.29	−0.002	0.00	−0.00	−0.00	<0.001

β, standardized coefficient; B, unstandardized coefficient indicating the expected unit change in the dependent variable for one unit change in the independent variable; SE, standard error; CI, confidence interval; * *n* = 77 except for models that include the Siconolfi step test and 6-min walk test (6MWT) (*n* = 49). All regression models were adjusted for age, SLEDAI, SDI, and disease duration. Handgrip was additionally adjusted for body weight.

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
