# Peer review of "Physical Fitness and Body Composition in Women with Systemic Lupus Erythematosus"

_medicina, 2019, doi:10.3390/medicina55020057_

Reviewer 1 Report

ABSTRACT

Remove the word “cross-sectional” in line 15. This is the “Background” section. The specification of the dataset-type is more relevant to the “Methods” section of the Abstract. For example, you can add the word “cross-sectional” in lines 18-19 to make the sentence read “A total of 77 women with SLE…during the previous 6 months were included in this cross-sectional study”.

INTRODUCTION

Although this section is written well in terms of the use of English language, flow of the concepts through neat ordering of the sentences within paragraphs, and neatly ordered paragraphs, there is a problem with the section in general in terms of the goal of a typical Introduction section: It somehow leaves the reader feel no motivation for the study aim. The literature review is mostly focused on general population, and not on women with SLE as much. The literature on the general population and the SLE population are not separated well to emphasize the latter. The section falls short of conveying the desired or intended message of the authors that the study constitutes a true contribution to the field in parallel to its aim.

Secondly, the section sets a tone as if authors are interested in the effects of physical fitness on body composition, thereby, cardiovascular health ultimately, on which physical fitness may exert a direct and an indirect effect through body composition. But then, authors regressions, although authors are honest in recognizing that their analyses’ results may not be interpreted causally and are using the correct wording while describing their results, seem to have “physical fitness” as the dependent and “body composition” and the independent variable. Authors should use a consistent language in the Introduction to parallel what has been done empirically and analytically.

MATERIALS AND METHODS

Design and Participants

In line 76, delete the words “according to” and instead use “provided by”, which will enable you to delete the word “criteria” in line 77 so to avoid the repetitive use of the word “criteria” twice within the same sentence. Delete the word “with” in line 77. Insert the definition of clinical stability in line 77 following the phrase “clinical stability” to make the sentence read: “Inclusion criteria were: women from 18 to 60 years with ≥4 SLE classification criteria provided by the American College of Rheumatology [21], a minimum follow-up of one year at our unit, and clinical stability (i.e., the absence of changes in the systematic lupus erythematosus disease activity index (SLEDAI) and/or treatment) during the past 6 months of the study.

In line 81, place a coma, “,” after the words “last year”. In the same line, delete the words “to be” preceding the gerund verb “receiving”.

The authors must move first paragraphs under the “Results” section (i.e., lines 157 through 165) to the “Design and Participants” section by integrating it in “Design and Participants”, as appropriate. Currently, the lines 157 through 165 are strictly misplaced under the “Results”.

                Anthropometric Measures

In line 87, delete the word “and” following the parenthesis. Then, in line 88, insert the word “and” before the words “fat mass” preceded by coma, so for the relevant part of the sentence to be as in “……, and fat mass and lean mass with……”

In line 88, please add the words “in kg” following the words “lean mass”. If BMI is to be defined in parenthesis, which is a good habit to do for general reader audience, it would be more complete if it is defined as in “weight in kg/height in m2”. Accordingly, in line 89, please replace the parenthesis content of “(kg/m2)) with the one suggested above as in “(weight in kg/height in m2)”.

                Physical Fitness

Add the word “Measures” to this section’s title so that it appears as in “2.3. Physical Fitness Measures”

In line 100, use the word “when” in place of the word “if”.

In the first paragraph, the word “frequency” appears three times in lines 98, 102, and 104. Although one understand what authors mean, the better choice of word could be “rate” instead of “frequency”.

In line 108, place a coma, “,”, followed by the words “for stage 2”.

Unless the journal has particular rules on number of decimal digits to be used in numbers quoted within the text, then I recommend using no more than two decimal digits in the authors’ particular subject matter. In fact, even reporting to the one-decimal digit may suffice. So, better to use 16.29 instead of 16.287, etc. in line 108 and in other lines that have similar numerical content.

Merge line 108 into line 107 with the equation by placing a coma right after the end of the equation followed by the content of lines 108 and 109 as in “EQUATION, where……”

In line 112, I recommend rephrasing of the first sentence as follows: “The test was performed along a 50-meter circuit, which was broken into 5-meter long sections by 10 cones.” Please note that one can omit the words “which was” before the word “broken”.

I recommend rephrasing the second sentence that starts in line 112 and continues in line 113 as follows “The total distance was calculated as the number of complete laps plus the number of sections covered within the last lap in case of an incomplete final lap at the expiration of the allowed time.”

Remove the paragraph indentation in line 119 by moving the content of the line into the previous line, line 118.

Rephrase the sentence in lines 120 and 121, as follows: ”Alternating between the two hands, participants performed the test twice with one-minute break between the two attempts of each hand.”

 Rephrase the sentence in line 122 as follows: The best score of each hand was used to compute an average of the two scores.

 Last paragraph in lines 123 and 130 is all written in third-person, “the subject”. However, in line 126, the authors included the word “your”, which is a pronoun for the second person. Thus, the word “your” in line 126 must be removed. In order to be consistent with the rest of the paragraph, the words “the subject’s” can be used, instead.

                Other Measurements

In line 134, add a coma, “,”, after the parenthesis.

                Statistical Analysis

Move the section title to the next page so that it does not stand alone in a different page from the paragraph to which it belongs. This fix may not be needed after the whole lining of the manuscript is altered based on the preceding revisions suggested by this reviewer.

In line 143, convert the word “distribution” that is currently singular into its plural form as in ”distributions” while replacing the verb “was” in line 144 with the verb “were” for grammatical accuracy in subject-verb consistency.

Although the authors’ phrasing in the Introduction section was suggestive that the dependent variable would be body composition and the independent variable would be physical fitness, the readers will only find out it is the opposite once they reach the Statistical Analysis section, which is too tardy or delayed to divulge this information, even though the regression analyses performed are meant to establish only associations and not causation, in turn, making the direction of causation less relevant. This comment was previously brought up in the Introduction section, to which it also relates, by this reviewer. This reviewer has no particular suggestion as to what to alter. This will depend on the authors’ thoughts and hypotheses of the direction of causality, if causality may exist. This reviewer does not have a suggested revision or fix. However, this inconsistency should be remedied in some way deemed appropriate by the authors. 

In line 150, place a coma, “,” after the abbreviation SDI and before the words “and”.

RESULTS

As previously brought up by this reviewer, the first paragraph extending between the lines 157 and 165 must be moved into the “Design and Participants” sub-section included within the “Materials and Methods” section.  Therefore, Figure 1 should and will be part of the “Design and Participants” as well.

In lines 178 through 180, authors mention Figure 2 and its content. However, they fail to provide briefly (i.e., in a sentence or two) the conclusion derived from Figure 2 graphs. Please include a brief summary of the relationship (i.e., raw/crude/unadjusted associations) between the various physical fitness measures and body composition measures that is depicted in Figure 2. If the conclusion obtained from Figure 2 is deemed to be unimportant, thereby, is omitted from the text by the authors on purpose, then Figure 2 itself should also be omitted from the paper. Please note that this reviewer does not suggest removal or keeping of Figure 2 as part of the paper, but is rather making an argument of consistency in the paper.

An important comment on Figure 2 depictions: Authors explicitly state under the “Statistical Analysis” section that physical fitness measures are used as dependent variables and body composition measures are used as independent variables in the linear regression analyses. Customarily, thus most commonly, dependent variables are graphed along the y-axis (i.e., vertical axis) and independent variables are shown along the x-axis (i.e., horizontal axis) in a two-dimensional Cartesian planes such as the ones contained in Figure 2. However, authors’ graphs within Figure 2 are drawn to have the dependent variables (physical fitness measures) on the x-axis and the independent variables (body composition measures) on the y-axis, in contradiction to what they communicated to the readers in the “Statistical Analysis” section as for what their dependent variables are and what their independent variables are, which, on itself, was in contrast to the implications based on wording of the “Introduction” section. This revision suggestion is held by this reviewer even though these figures are showing raw/crude/unadjusted associations, if any, between physical fitness and body composition. It still is important to follow certain customary actions in graphical representations of our findings.

The paragraph included under the “Results” section that extends between the lines 184 and 206 requires major revisions in the use of language that must be implemented so that the authors’ findings are communicated accurately or make sense. This reviewer does not have time to revise each similar error observed throughout the paragraph, in spite of his/her intention and effort to provide a review that is as detailed, and thus helpful, as possible to the authors in improving several aspects of their manuscript for publication. However, this reviewer will provide two examples of the major error in authors’ communication of the findings on the associations obtained from the linear regression analyses.

Example 1:

Sentence starting towards the end of line 193:

“Higher muscular strength,…….., was associated with lower BMI….”

A suggested revision/fix is as follows:

“Muscular strength.....was inversely associated with BMI”. Instead of the word “inversely”, it is also possible to use the word “negatively”. There are multiple ways to revising the sentence. For example, an alternative could be: “There is an inverse relationship between muscular strength and BMI…..”.

Example 2 (similar to example 1):

Sentence starting in line 200:

“Finally, higher flexibility,…..., was associated with lower BMI…..”

A suggested revision/fix is as follows:

“Finally, flexibility,……, was inversely related to BMI…..”.

In addition, this reviewer encourages the authors’ to take advantage of professional services of copy editors who are specialized in English language editing for journal articles, if needed. Having stated this recommendation, the sections prior to the “Results” section, interestingly, do not contain beyond minor errors related to the use of English language, overall. However, in the “Results” and the “Discussion” sections, the text contains major errors in the language use to the extent where certain sentences do not make any sense or are unclear or are confusing and vulnerable to several interpretations.

Table 2 requires multiple formatting changes. Firstly, although authors used American way/method of denoting decimal digits (i.e., decimal digits are followed after a period sign, “.”) vs. European way/method of denoting decimal digits (i.e., decimal digits are followed after a coma, “,”) within the text, they switched to European way in Table 2 where the decimals are shown/denoted with the use of “comas”. Authors must pick one way or the other by consulting the journal’s guidelines and use that way/method of denoting decimals consistently throughout the manuscript, including the text, figures, and tables, as relevant.

Similar comment previously brought up in regards to Figure 2 related to the display of dependent and independent variables on x-axis and y-axis, respectively, applies to Table 2 as well. In that, in Table 2, authors chose to show in bold the measures used as independent variables and listed the dependent variables underneath each dependent variable. This reviewer suggests that this is not a typical practice in Economics, Public Health, Statistics, Epidemiology, or in Medical journals. I recommend revising Table 2 by dividing the table into panels, where each panel is a physical fitness measure in which the magnitude and sign of the associations with body composition measures as obtained from the linear regression analyses’ coefficients are shown, instead of the current setting where each panel is a body composition measure in which the magnitude and sign of the associations with physical fitness measures as obtained from the regression analyses’ coefficients are shown.

In Table 2, there is a column titled “Beta” and there is a column titled “B”. Neither in the “Results” section nor in any other place in the paper authors explain what “B” is or represents (i.e., what the numbers reported in the column titled “B” in Table 2 mean and which analyses they are obtained from). This is inappropriate. Authors must correct this by including an explanation for the column titled “Beta” in the “Results” section, followed by a brief summary of the findings/numbers reported in the column titled “Beta” in Table 2. Although authors do not mention explicitly, it is clarified from the parenthesis included in the “Results” section that follow the explanations for the linear regression analyses’ findings and contained the numerical values of the coefficients/CIs, etc. shown in Table 2, it was possible to figure out what the numbers in column titled “B” in Table 2 were and from what analyses they were obtained.

Once again, I encourage authors to report no more than at the most two decimal digits in their numerical findings shown in the text, tables, and/or figures, as applicable.  The reason is two folds: 1) the level of precision implied by the third decimal digit shown in Table 2 is not needed (i.e., findings are not that precise for the third decimal digit to matter) and 2) for easier legibility of numerical findings reported, not to mention ink and paper space saved for environmental reasons.

Based on the third decimal digits shown in Table 2, this review caught rounding errors when these Table 2 numerical findings were stated within the text under the “Results” section in parenthesis in two decimal digits. There are more than one such rounding error present in the text. One example is in lines 186 and 187: “CI=-2.35 to -2.71” should have been “CI=2.36 to -1.31;……” when 2.358 is rounded to the two decimal digits and when 1.308 is rounded to the two decimal digits. Please proof-read the entire text to identify and correct such rounding errors.

In line 221, a coma, “,”, should be placed before the word “and”. Although this author has avoided the mention of all similar errors of omitted comas that exist in other sentences throughout the text in the interest of time, this does not mean there are no other such errors. Indeed, such punctuation related errors are not limited to the few examples this reviewer has brought up in this review.

In Figure 3, please remove the repetition of the units or measure units following the numbers on the y-axis in each graph. Instead, authors can improve the legibility and aesthetics of these Figure 3 graphs by including the y-axis unit of measure right next to the sideway displayed y-axis titles. For example, the first graph in Figure 3 depicts BMI on the y-axis and thus the sideway displayed axis title is “BMI”. Authors can, instead, print “BMI in kg/m2” or “BMI (kg/m2)” as for the y-axis title. While this revision will explicitly communicate to the reader that all numbers printed on the y-axis has the same unit of measure of BMI, which is kg/m2, it will also provide for better aesthetics and easier legibility of the numbers printed on the y-axis. The same logic applies for all other graphs of Figure 3.

DISCUSSION

This section also contains errors in the use of English language, as mentioned in prior comments above and will not be repeated here.

Conceptually, the section does not demonstrate an ideal example of discussing findings. Part of this criticism (provided in hopes of constituting constructive criticism and not the other way around) is residing in the unfavorable use of the English language, part is due to less than ideal organization/ordering of the sentences within the section, and yet and last but not least, another reason is not to do with the use of English language. An example of the latter reason could be found in line 279. The sentence in line 279 has two parts. The first part is clear whereas the second is not. What is meant by “lack of association across BMI categories”? What are BMI categories? Do authors refer to “weight status categories” typically used in the literature such as underweight, normal weight, overweight, obese type I, obese II, etc.? If I assume the answer is yes, then if grip strength is found in that study to be positively related to BMI, then one would expect weight status categories indicating higher weight to height ratio (i.e., BMI) to be associated with the grip strength. Is not this the case? If it is the case, it would be more meaningful to revise the authors’ summary of the study 39 along these lines. Otherwise, the sentence is far from communicating any point in part, the second part, which also reduces the reliability of the first part.

In reference to the same sentence provided in line 279: conditional on the accuracy of my interpretation of what authors meant in the whole sentence, a potential explanation of the finding in study 39 (I am not familiar with the study which I should disclose at once, so my opinion is solely based on what the authors’ of the present study cited from reference study 39 in line 279, and, in addition, this is only to the extent I interpreted the authors’ sentence correctly) could be due to the fact that BMI is not a perfect measure of healthy/unhealthy weight for a given height. This is told to be especially true for people, such as athletes, with high muscle mass. Muscle, on average, has higher density than fat. Thus, muscly people, albeit they may appear as normal weight and are at a healthy weight, may turn out with BMI above 25 indicating at least a status of overweight. If grip strength is positively correlated with higher muscle mass/weight, then muscly people may have higher grip strength scores and also have higher BMI due to muscles weighting heavier, which may be a factor that may explain the positive association found by study 39 between BMI and grip strength. However, if muscly people’s BMI is overstated due to the higher density of muscles than fat resulting in a BMI of at least 25 but less than 30, then such people will be classified as overweight based solely on BMI as a measure of healthy/unhealthy weight status. At the same time, if such muscly people tend to be less likely to have BMI 30 or above, they will be less likely to be in obese status. In such a case, the positive association between BMI and grip strength only will prevail for overweight category based on BMI ranges, but not for obese categories with BMI of 30 or more. Hence, a lack of association between grip strength and BMI categories may happen for BMI scores 30 or above, while a positive association may exist for BMI scores lower than 25, etc. This may potentially constitute an explanation for the positive association appearing for some BMI categories for women but not for all BMI categories as found in study 39 per what the authors of the present study wrote provided I understood what the authors of the present study meant correctly.

In reference to the sentences in lines 274 through 277: firstly, I would avoid including both BMI and weight as independent variables simultaneously within the same regression. Secondly and as consequent of the first comment, I will not attach any interpretation to the change in findings when weight is added alongside BMI as an independent variable within the same regression. I recommend removing BMI as one of the independent variables and instead using weight and height as two separate independent variables and compare the findings obtained from the regression with BMI as the independent variable vs. the regression with weight and height as two separate independent variables, at the most.

The sentence that starts in line 277 and continues into line 278 is unclear mainly because it is not grammatically accurate. Please revise.

In line 284, it is not clear to what/which test authors refer when saying “….,and this test”. In general, this sentence would benefit from rephrasing.

In line 290, please replace the word “increasing” with “increased”.

In line 297, if it is interesting to examine whether muscle strength can counteract the effect of obesity for SLE patients”, then authors should provide a theory-based hypothesis (i.e., based on biology and sub-disciplines, etc.) as to why SLE patients may be or may not be expected to exhibit the same relationship that has been shown to exist between obesity and muscle strength in the general population. In fact, it is because of the lack of a thorough discussion of potential mechanisms that may relate/associate physical fitness and body composition in women with SLE in a systematic and well-organized way based on theories and previous literature’s findings, as available and related, the Introduction and the Discussion sections of this manuscript are deemed to have written in a less than favorable way and do fall short of communicating the motivation for and the contribution of this study.

Another example of a poorly constructed sentence, which, thereby, lacks clarity is the sentence starting towards the end of line 304 and roughly ending in the middle of line 306. Again, the less than ideally constructed sentences that are not clear in meaning are not limited to what have been listed by this reviewer within this section and across the sections of the manuscript. This reviewer did its best to provide the most and the more important ones of the examples of such sentences to be able to help authors improve the manuscript for publication.

In line 256, please remove the word “concurring” and use the word “consistent”, in place of the removed word. In the same line, replace the proposition “in” with the proposition “for”.

In line 267, please replace the word/verb “is” with its past tense form “was” for consistency of tense within the section. In general, please proof-read the whole manuscript for consistency of use in tense, past vs. present. This reviewer may have missed such inconsistency and omitted the ones caught in the interest of time, etc. In the same line, please delete the word “lower”. It is not needed. It is implied.

The sentence in lines 277-278 is unclear in terms of its meaning and this is partially due to being incorrectly constructed, a grammar error.

The study limitations are not comprehensive. The presence of longitudinal data may or may not suffice to establish causality. It depends. The authors’ findings may not even be generalizable to any population including SLE patients of the told hospitals at that period of time. This reviewer does not understand what it is meant by “reasonable number of confounders”. It is not about the number of confounders but it is about the quality of research design that takes care of the suggested mechanisms through which confounding of the findings may occur in a study. Thus, this reviewer does not find the sentence in line 319 to be consistent with scientific writing. Since authors did not provide in any section a thorough and clear presentation of the mechanisms and pathways through which physical fitness and body composition may relate specifically for SLE patients, there is no way to know for a non-medical professional to judge whether the control co-variates used in the study were fine, redundant, needed, and/or yielded collinearity issues.  

CONCLUSION

Please rephrase the first sentence in lines 324 and 325, parallel to the examples provided under the “Results” section above by this reviewer.

Place a coma, “,”, in line 326 followed by the words “muscle strength”.  

Based on the overall feedback provided to this point, I also recommend revision to the final sentence of this section starting at the end of line 328 and continuing until line 331.

This reviewer congratulates the authors for having finished their manuscript and submitted it to the Medicina journal. In addition, this reviewer hopes that the feedback provided above will prove helpful for the authors of the study while revising and improving he current version of their manuscript. It will be a pleasure to see this paper published in Medicina or another journal soon.

Based on the version reviewed by this reviewer, however, the paper is not yet ready to be published as is. Therefore, the recommendation of this reviewer to the Editors of the Journal is “resubmit after major revisions”.

Author Response

Reviewer 1

Reviewer comment: ABSTRACT. Remove the word “cross-sectional” in line 15. This is the “Background” section. The specification of the dataset-type is more relevant to the “Methods” section of the Abstract. For example, you can add the word “cross-sectional” in lines 18-19 to make the sentence read “A total of 77 women with SLE…during the previous 6 months were included in this cross-sectional study”.

Author’s response: We have corrected the abstract following the reviewer’s suggestion (page 1, line 20).

Reviewer comment:
INTRODUCTION. Although this section is written well in terms of the use of English language, flow of the concepts through neat ordering of the sentences within paragraphs, and neatly ordered paragraphs, there is a problem with the section in general in terms of the goal of a typical Introduction section: It somehow leaves the reader feel no motivation for the study aim. The literature review is mostly focused on general population, and not on women with SLE as much. The literature on the general population and the SLE population are not separated well to emphasize the latter. The section falls short of conveying the desired or intended message of the authors that the study constitutes a true contribution to the field in parallel to its aim.

Author’s response: Thank you for arising this important issue. The reviewer is absolutely right. We have now clearly distinguished between general population and patients with SLE and emphasized on the relevance of the study and novelty of the data presented (page 2, lines 60-73).

Reviewer comment: Design and Participants. In line 76, delete the words “according to” and instead use “provided by”, which will enable you to delete the word “criteria” in line 77 so to avoid the repetitive use of the word “criteria” twice within the same sentence. Delete the word “with” in line 77. Insert the definition of clinical stability in line 77 following the phrase “clinical stability” to make the sentence read: “Inclusion criteria were: women from 18 to 60 years with ≥4 SLE classification criteria provided by the American College of Rheumatology [21], a minimum follow-up of one year at our unit, and clinical stability (i.e., the absence of changes in the systematic lupus erythematosus disease activity index (SLEDAI) and/or treatment) during the past 6 months of the study.

Author’s response: The comments are highly appreciated and we have included changes according to them (page 2, line 82).

Reviewer comment: In line 81, place a coma, “,” after the words “last year”. In the same line, delete the words “to be” preceding the gerund verb “receiving”.

Author’s response: Done, Thanks.

Reviewer comment:
The authors must move first paragraphs under the “Results” section (i.e., lines 157 through 165) to the “Design and Participants” section by integrating it in “Design and Participants”, as appropriate. Currently, the lines 157 through 165 are strictly misplaced under the “Results”.

Author’s response: Done. Thank you for the suggestion.

Reviewer comment: Anthropometric Measures. In line 87, delete the word “and” following the parenthesis. Then, in line 88, insert the word “and” before the words “fat mass” preceded by coma, so for the relevant part of the sentence to be as in “……, and fat mass and lean mass with……”

Author’s response: The comments are highly appreciated and we have included changes according to them.

Reviewer comment: In line 88, please add the words “in kg” following the words “lean mass”. If BMI is to be defined in parenthesis, which is a good habit to do for general reader audience, it would be more complete if it is defined as in “weight in kg/height in m2”. Accordingly, in line 89, please replace the parenthesis content of “(kg/m2)) with the one suggested above as in “(weight in kg/height in m2)”.

Author’s response: The comments are highly appreciated and we have added this information in page 2, line 106.

Reviewer comment:
Physical Fitness. Add the word “Measures” to this section’s title so that it appears as in “2.3. Physical Fitness Measures”

Author’s response: The comments are highly appreciated and we have included this information in the section’s title.

Reviewer comment: In line 100, use the word “when” in place of the word “if”.

Author’s response: Thanks, we have included the changes according to the suggestion.

Reviewer comment: In the first paragraph, the word “frequency” appears three times in lines 98, 102, and 104. Although one understand what authors mean, the better choice of word could be “rate” instead of “frequency”.

Author’s response: We have changed “frequency” by “rate”.

Reviewer comment: In line 108, place a coma, “,”, followed by the words “for stage 2”.

Author’s response: Thank you for the suggestion.

Reviewer comment:
Unless the journal has particular rules on number of decimal digits to be used in numbers quoted within the text, then I recommend using no more than two decimal digits in the authors’ particular subject matter. In fact, even reporting to the one-decimal digit may suffice. So, better to use 16.29 instead of 16.287, etc. in line 108 and in other lines that have similar numerical content.

Author’s response: Please note that the values mentioned by the reviewer do not refer to the authors study results but to the original formulas by Siconolfi et al. (1985). We respectfully do not feel we should change this nor reduce the number of decimal points in this case in order to be consistent with the original source.

Reviewer comment: Merge line 108 into line 107 with the equation by placing a coma right after the end of the equation followed by the content of lines 108 and 109 as in “EQUATION, where……”

Author’s response: We have modified the text according to the reviewer’s suggestion. Thanks.

Reviewer comment:
In line 112, I recommend rephrasing of the first sentence as follows: “The test was performed along a 50-meter circuit, which was broken into 5-meter long sections by 10 cones.” Please note that one can omit the words “which was” before the word “broken”.

Author’s response: Thank you very much, we have included the changes according to the suggestion in line 132.

Reviewer comment: I recommend rephrasing the second sentence that starts in line 112 and continues in line 113 as follows “The total distance was calculated as the number of complete laps plus the number of sections covered within the last lap in case of an incomplete final lap at the expiration of the allowed time.”

Author’s response: Thank you very much for the suggestion. We have included this information in lines 133-135.

Reviewer comment:
Remove the paragraph indentation in line 119 by moving the content of the line into the previous line, line 118.

Author’s response: We have included this change according to the reviewer’s comment.

Reviewer comment: Rephrase the sentence in lines 120 and 121, as follows: ”Alternating between the two hands, participants performed the test twice with one-minute break between the two attempts of each hand.”

Author’s response: Thank you very much for the suggestion. We have included this information in lines 140-141.

Reviewer comment:
Rephrase the sentence in line 122 as follows: The best score of each hand was used to compute an average of the two scores.

Author’s response: Thank you very much for the suggestion. We have included this information in lines 141-142.

Reviewer comment: Last paragraph in lines 123 and 130 is all written in third-person, “the subject”. However, in line 126, the authors included the word “your”, which is a pronoun for the second person. Thus, the word “your” in line 126 must be removed. In order to be consistent with the rest of the paragraph, the words “the subject’s” can be used, instead.

Author’s response: The comments are highly appreciated. We have included this change in lines 143-151

Reviewer comment:
Other Measurements. In line 134, add a coma, “,”, after the parenthesis.

Author’s response: Done. Thanks.

Reviewer comment: Statistical Analysis. Move the section title to the next page so that it does not stand alone in a different page from the paragraph to which it belongs. This fix may not be needed after the whole lining of the manuscript is altered based on the preceding revisions suggested by this reviewer.

Author’s response: Thanks.

Reviewer comment:
In line 143, convert the word “distribution” that is currently singular into its plural form as in ”distributions” while replacing the verb “was” in line 144 with the verb “were” for grammatical accuracy in subject-verb consistency.

Author’s response: Thank you very much for the suggestion. We have included this changes in line 166.

Reviewer comment: Although the authors’ phrasing in the Introduction section was suggestive that the dependent variable would be body composition and the independent variable would be physical fitness, the readers will only find out it is the opposite once they reach the Statistical Analysis section, which is too tardy or delayed to divulge this information, even though the regression analyses performed are meant to establish only associations and not causation, in turn, making the direction of causation less relevant. This comment was previously brought up in the Introduction section, to which it also relates, by this reviewer. This reviewer has no particular suggestion as to what to alter. This will depend on the authors’ thoughts and hypotheses of the direction of causality, if causality may exist. This reviewer does not have a suggested revision or fix. However, this inconsistency should be remedied in some way deemed appropriate by the authors.

Author’s response: Thank you very much for catching this relevant issue. As stated by the reviewer, the introduction and statistical analysis sections were not consistent in relation to the specification of the dependent and the independent variables. This was because of an error in the statistical analysis section and we have corrected it to express that body composition and physical fitness were the dependent and independent variables in all regression models, respectively. (page 5, line 169-171).

Reviewer comment:
In line 150, place a coma, “,” after the abbreviation SDI and before the words “and”.

Author’s response: The comment is highly appreciated and we have included this change in line 171.

Reviewer comment: RESULTS. As previously brought up by this reviewer, the first paragraph extending between the lines 157 and 165 must be moved into the “Design and Participants” sub-section included within the “Materials and Methods” section.  Therefore, Figure 1 should and will be part of the “Design and Participants” as well.

Author’s response: Done, Thanks. (page 2, line 102).

Reviewer comment:
In lines 178 through 180, authors mention Figure 2 and its content. However, they fail to provide briefly (i.e., in a sentence or two) the conclusion derived from Figure 2 graphs. Please include a brief summary of the relationship (i.e., raw/crude/unadjusted associations) between the various physical fitness measures and body composition measures that is depicted in Figure 2. If the conclusion obtained from Figure 2 is deemed to be unimportant, thereby, is omitted from the text by the authors on purpose, then Figure 2 itself should also be omitted from the paper. Please note that this reviewer does not suggest removal or keeping of Figure 2 as part of the paper, but is rather making an argument of consistency in the paper.

Author’s response: A brief summary of the raw association between physical fitness and body composition has been added to the text (page 6, in lines 189-195).

Reviewer comment: An important comment on Figure 2 depictions: Authors explicitly state under the “Statistical Analysis” section that physical fitness measures are used as dependent variables and body composition measures are used as independent variables in the linear regression analyses. Customarily, thus most commonly, dependent variables are graphed along the y-axis (i.e., vertical axis) and independent variables are shown along the x-axis (i.e., horizontal axis) in a two-dimensional Cartesian planes such as the ones contained in Figure 2. However, authors’ graphs within Figure 2 are drawn to have the dependent variables (physical fitness measures) on the x-axis and the independent variables (body composition measures) on the y-axis, in contradiction to what they communicated to the readers in the “Statistical Analysis” section as for what their dependent variables are and what their independent variables are, which, on itself, was in contrast to the implications based on wording of the “Introduction” section. This revision suggestion is held by this reviewer even though these figures are showing raw/crude/unadjusted associations, if any, between physical fitness and body composition. It still is important to follow certain customary actions in graphical representations of our findings.

Author’s response: Thank you for catching this error in the statistical analyses section where the dependent variables (body composition) and independent variables (physical fitness) were interchanged by error. This has been corrected to clarify that the dependent variables were those related to body composition and the independent one those related to fitness. With these modifications, Figure 2 now makes sense with the entire article.

Reviewer comment:
The paragraph included under the “Results” section that extends between the lines 184 and 206 requires major revisions in the use of language that must be implemented so that the authors’ findings are communicated accurately or make sense. This reviewer does not have time to revise each similar error observed throughout the paragraph, in spite of his/her intention and effort to provide a review that is as detailed, and thus helpful, as possible to the authors in improving several aspects of their manuscript for publication. However, this reviewer will provide two examples of the major error in authors’ communication of the findings on the associations obtained from the linear regression analyses.

Example 1:

Sentence starting towards the end of line 193:

“Higher muscular strength,…….., was associated with lower BMI….”

A suggested revision/fix is as follows:

“Muscular strength.....was inversely associated with BMI”. Instead of the word “inversely”, it is also possible to use the word “negatively”. There are multiple ways to revising the sentence. For example, an alternative could be: “There is an inverse relationship between muscular strength and BMI…..”.

Example 2 (similar to example 1):

Sentence starting in line 200:

“Finally, higher flexibility,…..., was associated with lower BMI…..”

A suggested revision/fix is as follows:

“Finally, flexibility,……, was inversely related to BMI…..”.

Author’s response: Following the reviewer’s suggestion, we did a major revision on the sections described by the reviewer (page 6, lines 203-219). We have carefully revised the English language along the entire “Results” section.

Reviewer comment: In addition, this reviewer encourages the authors’ to take advantage of professional services of copy editors who are specialized in English language editing for journal articles, if needed. Having stated this recommendation, the sections prior to the “Results” section, interestingly, do not contain beyond minor errors related to the use of English language, overall. However, in the “Results” and the “Discussion” sections, the text contains major errors in the language use to the extent where certain sentences do not make any sense or are unclear or are confusing and vulnerable to several interpretations.

Author’s response: We have undertaken careful revisions to the language and a Native speaker has revised the entire manuscript, with particular attention to “Results” and “Discussion” sections.

Reviewer comment:
Table 2 requires multiple formatting changes. Firstly, although authors used American way/method of denoting decimal digits (i.e., decimal digits are followed after a period sign, “.”) vs. European way/method of denoting decimal digits (i.e., decimal digits are followed after a coma, “,”) within the text, they switched to European way in Table 2 where the decimals are shown/denoted with the use of “comas”. Authors must pick one way or the other by consulting the journal’s guidelines and use that way/method of denoting decimals consistently throughout the manuscript, including the text, figures, and tables, as relevant.

Author’s response: Following the journal’s guidelines, Table 2 has been changed to American method of denoting decimal digit and the whole manuscript has been revised for consistency.

Reviewer comment: Similar comment previously brought up in regards to Figure 2 related to the display of dependent and independent variables on x-axis and y-axis, respectively, applies to Table 2 as well. In that, in Table 2, authors chose to show in bold the measures used as independent variables and listed the dependent variables underneath each dependent variable. This reviewer suggests that this is not a typical practice in Economics, Public Health, Statistics, Epidemiology, or in Medical journals. I recommend revising Table 2 by dividing the table into panels, where each panel is a physical fitness measure in which the magnitude and sign of the associations with body composition measures as obtained from the linear regression analyses’ coefficients are shown, instead of the current setting where each panel is a body composition measure in which the magnitude and sign of the associations with physical fitness measures as obtained from the regression analyses’ coefficients are shown.

Author’s response: Table 2 has been modified according to the reviewers recommendations. Thank you.

Reviewer comment:
In Table 2, there is a column titled “Beta” and there is a column titled “B”. Neither in the “Results” section nor in any other place in the paper authors explain what “B” is or represents (i.e., what the numbers reported in the column titled “B” in Table 2 mean and which analyses they are obtained from). This is inappropriate. Authors must correct this by including an explanation for the column titled “Beta” in the “Results” section, followed by a brief summary of the findings/numbers reported in the column titled “Beta” in Table 2. Although authors do not mention explicitly, it is clarified from the parenthesis included in the “Results” section that follow the explanations for the linear regression analyses’ findings and contained the numerical values of the coefficients/CIs, etc. shown in Table 2, it was possible to figure out what the numbers in column titled “B” in Table 2 were and from what analyses they were obtained.

Author’s response: Thank you for this suggestion. We have included further details of what B and Beta represent as well as more information about the analyses performed both in the text and as footnotes in Table 2.

Reviewer comment: Once again, I encourage authors to report no more than at the most two decimal digits in their numerical findings shown in the text, tables, and/or figures, as applicable.  The reason is two folds: 1) the level of precision implied by the third decimal digit shown in Table 2 is not needed (i.e., findings are not that precise for the third decimal digit to matter) and 2) for easier legibility of numerical findings reported, not to mention ink and paper space saved for environmental reasons.

Author’s response: The authors agree with the reviewer and we have deleted the third decimal digit from the Results section.

Reviewer comment:
Based on the third decimal digits shown in Table 2, this review caught rounding errors when these Table 2 numerical findings were stated within the text under the “Results” section in parenthesis in two decimal digits. There are more than one such rounding error present in the text. One example is in lines 186 and 187: “CI=-2.35 to -2.71” should have been “CI=2.36 to -1.31;……” when 2.358 is rounded to the two decimal digits and when 1.308 is rounded to the two decimal digits. Please proof-read the entire text to identify and correct such rounding errors.

Author’s response: We have identified and corrected every rounding error.

Reviewer comment: In line 221, a coma, “,”, should be placed before the word “and”. Although this author has avoided the mention of all similar errors of omitted comas that exist in other sentences throughout the text in the interest of time, this does not mean there are no other such errors. Indeed, such punctuation related errors are not limited to the few examples this reviewer has brought up in this review.

Author’s response: We have included a comma before “and” in the entire paper to improve the use of English.

Reviewer comment:
In Figure 3, please remove the repetition of the units or measure units following the numbers on the y-axis in each graph. Instead, authors can improve the legibility and aesthetics of these Figure 3 graphs by including the y-axis unit of measure right next to the sideway displayed y-axis titles. For example, the first graph in Figure 3 depicts BMI on the y-axis and thus the sideway displayed axis title is “BMI”. Authors can, instead, print “BMI in kg/m2” or “BMI (kg/m2)” as for the y-axis title. While this revision will explicitly communicate to the reader that all numbers printed on the y-axis has the same unit of measure of BMI, which is kg/m2, it will also provide for better aesthetics and easier legibility of the numbers printed on the y-axis. The same logic applies for all other graphs of Figure 3.

Author’s response: Following reviewer’s suggestion, Figure 3 has been changed.

Reviewer comment: DISCUSSION. This section also contains errors in the use of English language, as mentioned in prior comments above and will not be repeated here.

Author’s response: We have revised the Discussion section and we have improved the English language.

Reviewer comment: Conceptually, the section does not demonstrate an ideal example of discussing findings. Part of this criticism (provided in hopes of constituting constructive criticism and not the other way around) is residing in the unfavorable use of the English language, part is due to less than ideal organization/ordering of the sentences within the section, and yet and last but not least, another reason is not to do with the use of English language. An example of the latter reason could be found in line 279. The sentence in line 279 has two parts. The first part is clear whereas the second is not. What is meant by “lack of association across BMI categories”? What are BMI categories? Do authors refer to “weight status categories” typically used in the literature such as underweight, normal weight, overweight, obese type I, obese II, etc.? If I assume the answer is yes, then if grip strength is found in that study to be positively related to BMI, then one would expect weight status categories indicating higher weight to height ratio (i.e., BMI) to be associated with the grip strength. Is not this the case? If it is the case, it would be more meaningful to revise the authors’ summary of the study 39 along these lines. Otherwise, the sentence is far from communicating any point in part, the second part, which also reduces the reliability of the first part.

Author’s response: Thanks for the appreciation, we have rephrased the sentences (page 10, lines 298-301).

Reviewer comment:
In reference to the same sentence provided in line 279: conditional on the accuracy of my interpretation of what authors meant in the whole sentence, a potential explanation of the finding in study 39 (I am not familiar with the study which I should disclose at once, so my opinion is solely based on what the authors’ of the present study cited from reference study 39 in line 279, and, in addition, this is only to the extent I interpreted the authors’ sentence correctly) could be due to the fact that BMI is not a perfect measure of healthy/unhealthy weight for a given height. This is told to be especially true for people, such as athletes, with high muscle mass. Muscle, on average, has higher density than fat. Thus, muscly people, albeit they may appear as normal weight and are at a healthy weight, may turn out with BMI above 25 indicating at least a status of overweight. If grip strength is positively correlated with higher muscle mass/weight, then muscly people may have higher grip strength scores and also have higher BMI due to muscles weighting heavier, which may be a factor that may explain the positive association found by study 39 between BMI and grip strength. However, if muscly people’s BMI is overstated due to the higher density of muscles than fat resulting in a BMI of at least 25 but less than 30, then such people will be classified as overweight based solely on BMI as a measure of healthy/unhealthy weight status. At the same time, if such muscly people tend to be less likely to have BMI 30 or above, they will be less likely to be in obese status. In such a case, the positive association between BMI and grip strength only will prevail for overweight category based on BMI ranges, but not for obese categories with BMI of 30 or more. Hence, a lack of association between grip strength and BMI categories may happen for BMI scores 30 or above, while a positive association may exist for BMI scores lower than 25, etc. This may potentially constitute an explanation for the positive association appearing for some BMI categories for women but not for all BMI categories as found in study 39 per what the authors of the present study wrote provided I understood what the authors of the present study meant correctly.

Author’s response: Thank you very much for providing us with a possible interpretation of our results with respect to other studies. We have tried to clarify better the results of the study in the lines 297-304.

Reviewer comment: In reference to the sentences in lines 274 through 277: firstly, I would avoid including both BMI and weight as independent variables simultaneously within the same regression. Secondly and as consequent of the first comment, I will not attach any interpretation to the change in findings when weight is added alongside BMI as an independent variable within the same regression. I recommend removing BMI as one of the independent variables and instead using weight and height as two separate independent variables and compare the findings obtained from the regression with BMI as the independent variable vs. the regression with weight and height as two separate independent variables, at the most.

Author’s response: Please note that body composition variables were the dependent variables, as explained in previous answers. In this particular analysis, body weight was added to the model following the recommendation by Keevil et al. (ref 36) and Balogun et al. (ref 37), to ensure that body weight was taken into account when associations between BMI and handgrip strength were done. (page 10, lines 299-301)

Reviewer comment:
The sentence that starts in line 277 and continues into line 278 is unclear mainly because it is not grammatically accurate. Please revise.

Author’s response: The sentence has been rewritten (line 302-303). Thanks.

Reviewer comment: In line 284, it is not clear to what/which test authors refer when saying “….,and this test”. In general, this sentence would benefit from rephrasing.

Author’s response: The sentence has been rephrased (line 306).

Reviewer comment:
In line 290, please replace the word “increasing” with “increased”.

Author’s response: Done, Thanks.

Reviewer comment: In line 297, if it is interesting to examine whether muscle strength can counteract the effect of obesity for SLE patients”, then authors should provide a theory-based hypothesis (i.e., based on biology and sub-disciplines, etc.) as to why SLE patients may be or may not be expected to exhibit the same relationship that has been shown to exist between obesity and muscle strength in the general population. In fact, it is because of the lack of a thorough discussion of potential mechanisms that may relate/associate physical fitness and body composition in women with SLE in a systematic and well-organized way based on theories and previous literature’s findings, as available and related, the Introduction and the Discussion sections of this manuscript are deemed to have written in a less than favorable way and do fall short of communicating the motivation for and the contribution of this study.

Author’s response: A theory-based hypothesis regarding potential mechanisms in how muscle strength counteract the effect of obesity has been included in lines 312-316 following reviewer’s suggestion.

Reviewer comment:
Another example of a poorly constructed sentence, which, thereby, lacks clarity is the sentence starting towards the end of line 304 and roughly ending in the middle of line 306. Again, the less than ideally constructed sentences that are not clear in meaning are not limited to what have been listed by this reviewer within this section and across the sections of the manuscript. This reviewer did its best to provide the most and the more important ones of the examples of such sentences to be able to help authors improve the manuscript for publication.

Author’s response: We appreciate the constructive comments to improve the use of English throughout the text. The sentence has been modificated to improve clarity in lines 324-325.

Reviewer comment: In line 256, please remove the word “concurring” and use the word “consistent”, in place of the removed word. In the same line, replace the proposition “in” with the proposition “for”.

Author’s response: Done (line 274). Thanks.

Reviewer comment:
In line 267, please replace the word/verb “is” with its past tense form “was” for consistency of tense within the section. In general, please proof-read the whole manuscript for consistency of use in tense, past vs. present. This reviewer may have missed such inconsistency and omitted the ones caught in the interest of time, etc. In the same line, please delete the word “lower”. It is not needed. It is implied.

Author’s response: That line has been changed following the reviewer’s comment, and the entie manuscript has been carefully revised, as mentioned above. Thank you.

Reviewer comment: The sentence in lines 277-278 is unclear in terms of its meaning and this is partially due to being incorrectly constructed, a grammar error.

Author’s response: The sentence has been rewritten (line 302-303). Thanks.

Reviewer comment:
The study limitations are not comprehensive. The presence of longitudinal data may or may not suffice to establish causality. It depends. The authors’ findings may not even be generalizable to any population including SLE patients of the told hospitals at that period of time. This reviewer does not understand what it is meant by “reasonable number of confounders”. It is not about the number of confounders but it is about the quality of research design that takes care of the suggested mechanisms through which confounding of the findings may occur in a study. Thus, this reviewer does not find the sentence in line 319 to be consistent with scientific writing. Since authors did not provide in any section a thorough and clear presentation of the mechanisms and pathways through which physical fitness and body composition may relate specifically for SLE patients, there is no way to know for a non-medical professional to judge whether the control co-variates used in the study were fine, redundant, needed, and/or yielded collinearity issues. 

Author’s response: We have carefully revised and improved the limitations section by including more specific information and deleting other information that was not appropriate (i.e. that related to the confounders).

In addition, and not related to this section, we have made efforts to explain potential mechanisms related to physical fitness and body composition along the discussion.

Reviewer comment: CONCLUSION. Please rephrase the first sentence in lines 324 and 325, parallel to the examples provided under the “Results” section above by this reviewer.

Author’s response: Done. Thanks.

Reviewer comment:
Place a coma, “,”, in line 326 followed by the words “muscle strength”. 

Author’s response: The comment is highly appreciated and we have included this change.

Reviewer comment: Based on the overall feedback provided to this point, I also recommend revision to the final sentence of this section starting at the end of line 328 and continuing until line 331.

Author’s response: That sentence has been revised and improved in lines 350-351.

Reviewer comment:
This reviewer congratulates the authors for having finished their manuscript and submitted it to the Medicina journal. In addition, this reviewer hopes that the feedback provided above will prove helpful for the authors of the study while revising and improving he current version of their manuscript. It will be a pleasure to see this paper published in Medicina or another journal soon.

Author’s response: The authors would like to thank this reviewer for the comprehensive work on reviewing this manuscript. The extensive comments and suggestions have undoubtedly improved the quality of our work. We highly appreciate the time and effort that this takes. Thank you.

Reviewer 2 Report

This is a cross-sectional study investigating the association of fitness with body composition among 77 female systemic lupus erythematosus patients. The authors found that high fitness was associated with a lower BMI, fat mass index, and waist circumference.

Abstract

Please specify statistical analysis used to assess the association between fitness and body composition.

Text

Please specify characteristics of this study population. Were patients Caucasians in Spain?

Why did the authors not adjust for lifestyles including smoking? For example, smoking is associated with abdominal obesity and may confound the association between fitness and body composition. Explanations are conflicting between Statistical analysis section and Limitation section.

Table 1

It would be helpful if the authors could show the characteristics of participants according to the level of cardiorespiratory fitness (high vs. low).

Author Response

Reviewer 2

Reviewer comment:
Abstract. Please specify statistical analysis used to assess the association between fitness and body composition.

Author’s response: Thank you for catching this. Accordingly, we have included statistical analysis in the Abstract section (line 27).

Reviewer comment: Text. Please specify characteristics of this study population. Were patients Caucasians in Spain?

Author’s response: All patients were caucasians. More details regarding characteristics of this study population has been added in Design and Participants in line 79.

Reviewer comment: Why did the authors not adjust for lifestyles including smoking? For example, smoking is associated with abdominal obesity and may confound the association between fitness and body composition. Explanations are conflicting between Statistical analysis section and Limitation section.

Author’s response: We created an additional model including smoking status and the results were not altered. As we intended to keep a parisonious model, we decided not to include this variable in the final model presented. We have included more information regarding this in the methods section (page 5, lines 169-172).

Reviewer comment: Table 1. It would be helpful if the authors could show the characteristics of participants according to the level of cardiorespiratory fitness (high vs. low).

Author’s response: Thank you very much for the suggestion. We agree with the reviewer in the interest of comparing socio-demographic and clinical information between patients with high and low cardiorespiratory fitness. However, after a deep internal group discussion, we believe that this exceeds the aims of the present study. In addition, please note the sample size is relatively low and, particularly for cardiorespiratory fitness is even lower as the study was conducted in two waves and not all participants performed neither the Siconolfi nor the 6-minute walk tests. Therefore, we have respectfully decided not to present Table 1 divided by cardiorespiratory fitness levels. We hope that the reviewer understands our rationale.

Round  2

Reviewer 2 Report

Text

-       Why were cancer patients not excluded?

-       Specify whether p-values were two-sided or one-sided in the statistical analysis section.

-       The authors should modify the expression of p-values, following standard expression. Show exact p-values if above 0.001.

Table 1

-       I could not understand why mean and SD were calculated for categorical data (e.g., diabetes).

Figure 3

-       If data are unadjusted or crude mean and 95% CIs, specify so.

-       Minimum value is set as 0 for y axis. However, BMI, waist circumference, and other parameters usually do not have 0. The authors should reconsider range of y axis (especially lowest value).

Author Response

Reviewer comment: Text. Why were cancer patients not excluded?

Author’s response: The reviewer is correct and cancer was actually an exclusion criteria provided inflammation is commonly increased in such patients. Although none of the participants had cancer, we have included cancer as an exclusion criteria (page 2, line 86).

Reviewer comment: Text. Specify whether p-values were two-sided or one-sided in the statistical analysis section.

Author’s response: Our intention was to determine if there were differences between groups in both directions, so all p-values in our study were two-sided. Following this, we have added p-values information in statisstical analysis section. (page 5, line 177).

Reviewer comment: Text. The authors should modify the expression of p-values, following standard expression. Show exact p-values if above 0.001.

Author’s response: All p-values above 0.001 have been modified to show exact values. Thanks.

Reviewer comment: Table 1. I could not understand why mean and SD were calculated for categorical data (e.g., diabetes).

Author’s response: Thank you very much for catching this error. For categorical data, we are now showing percentages (Table 1).

Reviewer comment: Figure 3. If data are unadjusted or crude mean and 95% CIs, specify so.

Author’s response:  Data in Figure 3 was adjusted for age, disease activity, disease duration, and accrual damage. This information has been included into the text (lines 267-268).

Reviewer comment: Figure 3. Minimum value is set as 0 for y axis. However, BMI, waist circumference, and other parameters usually do not have 0. The authors should reconsider range of y axis (especially lowest value).

Author’s response: Following reviewer’s suggestion, the range of the y axis in Figure 3 has been modified.